# Divergent Asymmetric Total Synthesis of All Four Pestalotin Diastereomers from (*R*)-Glycidol

**DOI:** 10.3390/molecules25020394

**Published:** 2020-01-17

**Authors:** Mizuki Moriyama, Kohei Nakata, Tetsuya Fujiwara, Yoo Tanabe

**Affiliations:** Department of Chemistry, School of Science and Technology, Kwansei Gakuin University, 2-1 Gakuen, Sanda, Hyogo 669-1337, Japan; dbe04644@kwansei.ac.jp (M.M.); knakata01@okuno.co.jp (K.N.); tt_fujiwara@jp.daicel.com (T.F.)

**Keywords:** asymmetric total synthesis, divergent synthesis, pyran-2-one, pestalotin, epipestalotin, asymmetric Mukaiyama aldol reaction, hetero Diels-Alder reaction, Mitsunobu inversion, Chan’s diene, Brassard’s diene

## Abstract

All four chiral pestalotin diastereomers were synthesized in a straightforward and divergent manner from common (*R*)-glycidol. Catalytic asymmetric Mukaiyama aldol reactions of readily-available bis(TMSO)diene (Chan’s diene) with (*S*)-2-benzyloxyhexanal derived from (*R*)-glycidol produced a *syn*-aldol adduct with high diastereoselectivity and enantioselectivity using a Ti(*i*OPr)_4_/(*S*)-BINOL/LiCl catalyst. Diastereoselective Mukaiyama aldol reactions mediated by catalytic achiral Lewis acids directly produced not only a (1′*S*,6*S*)-pyrone precursor via the *syn*-aldol adduct using TiCl_4_, but also (1′*S*,6*R*)-pyrone precursor via the antialdol adduct using ZrCl_4_, in a stereocomplementary manner. A Hetero-Diels-Alder reaction of similarly available mono(TMSO)diene (Brassard’s diene) with (*S*)-2-benzyloxyhexanal produced the (1′*S*,6*S*)-pyrone precursor promoted by Eu(fod)_3_ and the (1′*S*,6*R*)-pyrone precursor Et_2_AlCl. Debenzylation of the (1′*S*,6*S*)-precursor and the (1′*S*,6*R*)-precursor furnished natural (−)-pestalotin (99% ee, 7 steps) and unnatural (+)-epipestalotin (99% ee, 7 steps), respectively. Mitsunobu inversions of the obtained (−)-pestalotin and (+)-epipestalotin successfully produced the unnatural (+)-pestalotin (99% ee, 9 steps) and (−)-epipestalotin (99% ee, 9 steps), respectively, in a divergent manner. All four of the obtained chiral pestalotin diastereomers possessed high chemical and optical purities (optical rotations, ^1^H-NMR, ^13^C-NMR, and HPLC measurements).

## 1. Introduction

Products possessing the 4-methoxy-5,6-dihydroxy-pyran-2-one structure are distributed in nature [1], including the (i) kavalactone series, such as kavain, methylsitan, dihydrokavain, dihydromethylsitan, etc. [2], and (ii) (−)-pestalotin [3], with the three unnatural diastereomers of (−)-epipestalotin, (+)-pestalotin, and (+)-epipestalotin (Figure 1). (−)-Pestalotin was isolated from *Pesalotia cryptomeriaecola* Sawada by Kimura and Tamura’s group; it possesses distinctive bioactivity as a gibberellin synergist [3,4,5]. Independently, the same compound was isolated from unidentified penicillium species as a minor component (code number: LLP-880α) by Ellestad’s group [6]. 

(−)-Pestalotin has received considerable attention as a synthetic target due to its characteristic structure, which includes two consecutive stereogenic centers. Several asymmetric total syntheses of (−)-pestalotin have therefore been performed to date, and the features are described in chronologic order of their development: (i) Dianion addition using ethyl acetoacetate with aldehyde containing a 1,3-dithian group, and successive asymmetric reduction using a chiral lithium hydro aluminate derived from chiral diamino tartrate, but with ca. 10% ee (Seebach’s group) [7]; (ii) Sharpless asymmetric kinetic resolution of allyl alcohol producing (−)-pestalotin and diastereomeric (−)-epipestalotin, and chiral pool synthesis starting from glycel aldehyde acetonide derived from d-mannitol to produce antipodal (+)-pestalotin and (+)-epipestalotin (Mori’s group) [8,9]; (iii) Derivatization of chiral diethyl tartarate and the incorporation of a tosyl group as a latent scaffold (Masaki’s group) [10]; (iv) Asymmetric reduction using (*S*)-alpine-borane reagent of ethynyl ketone intermediate and successive hetero-Diels-Alder reaction with Brassard’s siloxydiene [11] (Midland and Graham) [12]; (v) Chiral pool synthesis using unnatural (*S*)-norleucine, associated with successive syn-diastereoselective Mukaiyama aldol additions using Chan’s 1,3-disiloxydiene [13] (Hagiwara’s group) [14,15]; (vi) Cycloaddition strategy for chiral 1,2-diol with chiral induction utilizing Oppolzer’s camphor sultum (Curran and Zhang) [16]; (vii) Sharpless asymmetric dihydroxylation of ester including a non-conjugated ene-yne precursor (Wang and Shen) [17]; and (viii) Sharpless asymmetric dihydroxylation of ethyl heptenoate and successive β-ketoester formation via Birch reduction of the *m*-methoxyphenyl ring (Rao’s group) [18]. 

A review of these fruitful works revealed that the synthesis of all four pestalotin diastereomers is limited to the report by Mori’s group [9]. The syntheses are somewhat lengthy [(−)-pestalotin: 8 steps, 4% overall yield; (−)-epipestalotin: 6 steps, 9% overall yield; (+)-pestalotin: 10 steps, 1% overall yield; (+)-epipestalotin: 10 steps, 3% overall yield], and commence with two quite different starting compounds. Nonetheless, this work contributed significantly to clarifying the stereostructure-activity relationship of these families; 1′*S* configuration in the side chain was critical for the synergistic mode of action for gibberellin [6,9]. 

On the other hand, there are three natural 3-acyl-4-hydroxy-5,6-dihydroxy-pyran-2-one products relevant to 4-methoxy-5,6-dihydroxy-pyran-2-ones: (*R*)-podoblastins [19], (*R*)-lachnelluloic acid [20], and alternaric acid [21] (Figure 2). We previously reported asymmetric total syntheses of all these natural products utilizing a catalytic asymmetric Mukaiyama aldol reaction and an asymmetric Ti-Claisen condensation as the crucial steps [22,23]. 

Consistent with our expeditious total syntheses of all these compounds, we envisaged a divergent synthetic access to all four chiral pestalotin diastereomers starting from a common and readily-available chiral building block, i.e., (*R*)-glycidol.

## 2. Results and Discussion

### 2.1. General Strategy for the Total Syntheses of All Four Pestalotin Diastereomers

A couple of the present divergent strategies involve a catalytic asymmetric and a diastereoselective Mukaiyama aldol addition, and a diastereoselective hetero-Diels-Alder reaction, followed by a Mitsunobu inversion as the crucial steps (Scheme 1). (*R*)-Glycidol is transformed to a common starting (*S*)-2-benzyloxyhexanal (**1**) by the epoxide opening with a Grignard reagent. Syn- and anti-selective Mukaiyama aldol additions of readily-available bis(TMSO)diene (so-called Chan’s diene) **2** [13] with (*S*)-aldehyde **1** produce stereocomplementary chiral aldol adducts syn-**3** and anti-**3**, respectively. Alternatively, syn- and anti-selective hetero-Diels-Alder reactions of similarly available mono(TMSO)diene (so-called Brassard’s diene) **4** [11,24] with **1** produce diastereomeric chiral pyrone-adducts syn-**5** and anti-**5**, respectively. Following a conventional synthetic procedure [15], syn-**3** and anti-**3** are transformed to (−)-pestalotin and (+)-epipestalotin, respectively. Mitsunobu inversions of (−)-pestalotin and (+)-epipestalotin produce (−)-epipestalotin and (+)-pestalotin, respectively.

### 2.2. Total Syntheses of All Four Pestalotin Diastereomers

Synthesis of (*S*)-2-benzyloxyhexanal (**1**)

(*S*)-2-Benzyloxyhexanal (**1**) was synthesized from (*R*)-glycidol as shown in Scheme 2. (*R*)-Glycidol was converted to trityl ether **6** (or commercially available) as a crude solid, which was purified by recrystallization (83% yield). CuI-catalyzed Grignard reaction of *n*-PrMgBr with epoxide **6** [25] gave secondary alcohol **7** in 93% yield. After the benzyl group protection of **7**, the trityl group was removed using a PTS•H_2_O catalyst to afford primary alcohol **8** in 92% yield (2 steps). Finally, TEMPO (or Swern) oxidation of **8** produced (*S*)-2-benzyloxyhexanal **1** in 86% (or 97%) yield. Because of its easier recrystallization purification procedure, trityl protection method was selected instead of an alternative *p*-methoxybenzyl protective method. The present sequence (four steps and 61% overall yield) is superior regarding steps and overall yield compared with the relevant reported route starting from (*S*)-norleucine (five steps and 27% overall yield) [14].

### 2.3. Catalytic Asymmetric and Diastereoselective Mukaiyama Aldol Reactions 

With (*S*)-aldehyde **1** in hand, we next investigated a catalytic asymmetric Mukaiyama aldol reaction using readily-available Chan’s diene **2** [13] with **1** (Scheme 3). For this purpose, we employed the procedure applied for the asymmetric syntheses of (*R*)-podoblastin-S and (*R*)-lachnelluloic acid [22], as well as that described in Organic Syntheses, recently [26]. The reaction by using catalysis of Ti(*i*OPr)_4_ (2 mol%)/(*S*)-BINOL (2 mol%)/LiCl (4 mol%) and subsequent treatment with PPTS/MeOH afforded the desired aldol adduct syn-**3** in 31% yield with high diastereoselectivity and enantioselectivity [syn/anti = 93:7, 85% ee (C-5 position) by HPLC analysis].

Instead of (*S*)-BINOL, antipodal (*R*)-BINOL (6 mol %) was examined under identical conditions. Expectedly, the results differed with regard to the yield and diastereoselectivity [*syn*/*anti* = 50:50, 89% ee (*syn*), and 99% ee (*anti*) by HPLC analysis] (mismatching).

Pyrone formation and successive *O*-methylation using syn-**3** according to the reported method [15] produced 4-methoxy-5,6-dihydro-2*H*-pyran-2-one precursor (1’*S*,6*S*)-**5** in 88% yield (dr = 91:9) in two steps (Scheme 4). Finally, Pd/C-catalyzed debenzylation of (1′*S*,6*S*)-**5** furnished (−)-pestalotin in 60% yield and 99% ee (C-6 position) by HPLC analysis after recrystallization, together with a trace amount of (+)-epipestalotin.

### 2.4. Diastereoselective Mukaiyama Aldol Reactions Promoted by Achiral Lewis Acids 

Several simpler achiral Lewis acids were screened for diastereoselective Mukaiyama aldol reactions (Table 1). Hagiwara’s pioneering work addressed Lewis acid-mediated crossed-aldol reactions between **1** and **2** to afford syn-**3** adducts [15]; TiCl_4_ (100 mol %) produced excellent *syn*-**3** diastereoselectivity, but the anti-**3** selectivity was insufficient when using several other Lewis acids (BF_3_•OEt_2_, Et_2_AlCl, ZnCl_2_). Taking this information into account, we reinvestigated this procedure with the aim of enhancing stereocomplementary anti-**3** selectivity. The salient features are as follows: (i) The amount of TiCl_4_ (100 mol %) could be decreased to a catalytic amount (20 mol %), by which aldehyde **1** was sufficiently consumed (entries 1–3). (ii) Notably, the aldol-step reaction mixture was directly treated with PPTS/MeOH solution following the procedure mentioned described in Section 2.2 to furnish the desired 4-methoxy-5,6-dihydro-2*H*-pyran-2-one precursor (1′*S*,6*S*)-**5** smoothly with good syn-/anti- selectivity and excellent enantioselectivity at the C6-position (entry 2). This one-pot furan formation is the first finding among previously reported total syntheses. (iii) The use of other strong Lewis acids such as AlCl_3_, SnCl_4_, and BF_3_•OEt_2_, did not afford fruitful results (entries 4–6). (iv) Fortunately, the reaction using ZrCl_4_ switched the selectivity from syn- to anti- to afford (1′*S*,6*S*)-**5** as a major product with moderate diastereoselectivity but with excellent enantioselectivity (entry 8). (v) The use of mild metal triflate reagents such as M(OTf)n (M = Sc, La, Cu) were examined next. In contrast to TiCl_4_ and ZrCl_4_, Cu(OTf)_2_ produced a satisfactory yield with excellent syn-**3** selectivity and enantioselectivity (entry 11). 

### 2.5. Catalytic Diastereoselective Hetero-Diels-Alder Reaction 

Next, our attention was focused on a hetero-Diels-Alder reaction between aldehyde **1** and Brassard’s siloxydiene (**4**) [11] to construct pyrone precursors (1′*S*,6*S*)-**5** and (1′*S*,6*R*)-**5** in a straightforward manner, basically according to Midland’s protocol [12] (Scheme 2). The salient features are as follows: (i) Several Lewis acid catalysts (TiCl_4,_ AlCl_3_, SnCl_4_, BF_3_•OEt_2_, ZnCl_2_, and MgCl_2_) were screened (Table 2). The reaction profile apparently differed from the result listed in Table 1; i.e., both the yield and stereoselectivities were moderate to low (entries 1–5). (ii) Among metal triflate catalysts M(OTf)n (M = Sc, La, Cu), only Sc(OTf)_3_ afforded moderate result (entry 6), and, in contrast to our expectation Cu(OTf)_2_ afforded a disappointing result (entry 8). (iii) A reinvestigation of Midland’s best conditions using “chiral” Eu(hfc)_3_ revealed good selectivity for (1’*S*,6*S*)-**5** (entry 9). (iv) Notably, the use of more inexpensive and accessible “achiral” Eu(fod)_3_ produced superior diastereoselectivity and enantioselectivity (entry 10).

According to Midland’s report, stereocomplementary (1′*S*,6*R*)-diastereoselective reaction using Et_2_AlCl catalyst was examined to obtain pyrone (1′*S*,6*R*)-**5** in our hands (Scheme 5). Due to the subtle reported conditions, the reaction was hardly reproducible, and our best result was addressed; the obtained crude product contained considerable amounts of aldol-type compound **9** with the desirable product (1′*S*,6*R*)-**5**. Compound **9** was converted to (1′*S*,6*R*)-**5** by PPTS/toluene under reflux conditions, albeit in poor yield (12%). 

Finally, debenzylation of (1′*S*,6*S*)-**5** and (1’*S*,6*R*)-**5** using the H_2_/Pd(OH)_2_‒C catalyst produced (−)-pestalotin and (+)-epipestalotin, respectively, in good yield and with excellent optical purities (Scheme 6). Gratifyingly, Mitsunobu inversions of (−)-pestalotin and (+)-epipestalotin smoothly proceeded to furnish (+)-epipestalotin and (−)-pestalotin, respectively (Scheme 6). The present inversion step increases the value of the whole synthesis by a convergent process. Physical and spectral data (mp, optical rotation, ^1^H-NMR) of all four pestalotin diastereomers matched completely with Mori’s reported data [9]. Additional ^13^C-NMR spectral data and HPLC measurements are described in the experimental and in the ESI, respectively. The present divergent methodology is superior compared with Mori’s approach to the only reported total synthesis of all four pestalotin families [9] in the following respects: (i) common (*R*)-glycidol starting compound, (ii) short syntheses (7 and 9 steps), and (iii) higher total yield. 

## 3. Materials and Methods 

All reactions were carried out in oven-dried glassware under an argon atmosphere. Flash column chromatography was performed with silica gel 60 (230–400 mesh ASTM, Merck, Darmstadt, Germany). TLC analysis was performed on Merck 0.25 mm Silicagel 60 F_254_ plates. Melting points were determined on a hot stage microscope apparatus (ATM-01, AS ONE, Osaka, Japan) and were uncorrected. NMR spectra were recorded on a JEOLRESONANCE EXC-400 or ECX-500 spectrometer (JEOL, Akishima, Japan) operating at 400 MHz or 500 MHz for ^1^H-NMR, and 100 MHz and 125 MHz for ^13^C NMR. Chemical shifts (δ ppm) in CDCl_3_ were reported downfield from TMS (=0) for ^1^H-NMR. For ^13^C-NMR, chemical shifts were reported in the scale relative to CDCl_3_ (77.00 ppm) as an internal reference. Mass spectra were measured on a JMS-T100LC spectrometer (JEOL, Akishima, Japan). HPLC data were obtained on a SHIMADZU (Kyoto, Japan) HPLC system (consisting of the following: LC-20AT, CMB20A, CTO-20AC, and detector SPD-20A measured at 254 nm) using Chiracel AD-H or Ad-3 column (Daicel, Himeji, Japan, 25 cm) at 25 °C. Optical rotations were measured on a JASCO DIP-370 (Na lamp, 589 nm).

(*R*)-2-((trityloxy)methyl)oxirane (**6**)

TrCl (15.3 g, 55 mmol) in CH_2_Cl_2_ (35 mL) was added to a stirred solution of (*R*)-(+)-glycidol (3.70 g, 50 mmol) and Et_3_N (13.9 mL, 100 mmol) and DMAP (61 mg, 0.5 mmol) in CH_2_Cl_2_ (15 mL) at 0–5 °C under an Ar atmosphere, followed by stirring at 20–25 °C for 24 h. The mixture was quenched with sat. NH_4_Cl aq., which was extracted three times with Et_2_O. The combined organic phase was washed with water, brine, dried (Na_2_SO_4_), and concentrated. The obtained crude solid was purified by recrystallization from MeOH (100 mL) to give the desired product **6** (13.1 g, 83%).

Colorless crystals, mp 99–100 °C [lit. [25], 100 °C (EtOH)]; ^1^H-NMR (400 MHz, CDCl_3_): δ = 2.63 (dd, *J* = 2.3 Hz, 5.0 Hz, 1H), 2.78 (dd, *J* = 4.6, 1H), 3.09–3.18 (m, 2H), 3.32 (dd, *J* = 2.3 Hz, 10.0 Hz, 1H), 7.20–7.35 (m, 10H), 7.42–7.50 (m, 5H); ^13^C-NMR (100 MHz, CDCl_3_): δ = 44.6, 51.0, 64.7, 86.6, 127.0 (3C), 127.8 (6C), 128.6 (6C), 143.8. 


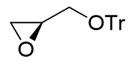


(*S*)-1-(Trityloxy)hexan-2-ol (**7**)

1-Bromopropane (8.60 mL, 95 mmol) was gradually added to a stirred Mg granular (2.31 g, 95 mmol) and a small amounts of I_2_ in THF (60 mL) at 20–25 °C under an Ar atmosphere, and the mixture was stirred for 0.5 h at 20–25 °C. CuI (143 mg, 0.80 mmol) was added, the mixture was cooled down to −40 °C and (*S*)-oxirane **6** (12.1 g, 38 mmol) in THF (100 mL) was added to the mixture at the same temperature, followed by stirring for 2 h. The mixture was quenched with sat. NH_4_Cl aq., which was extracted three times with AcOEt. The combined organic phase was washed with water, brine, dried (Na_2_SO_4_), and concentrated. The obtained crude product was purified by SiO_2_–column chromatography (hexane/AcOEt = 15/1) to give the desired alcohol **7** (12.7 g, 93%).

Pale yellow oil; ^1^H-NMR (400 MHz, CDCl_3_): δ = 0.86 (t, *J* = 6.9 Hz, 3H), 1.16–1.46 (m, 6H), 2.30 (d, *J* = 3.7 Hz, 1H), 3.02 (dd, *J* = 7.8 Hz, 9.2 Hz, 1H), 3.18 (dd, *J* = 3.2 Hz, 9.2 Hz, 1H), 3.72–3.80 (m, 1H), 7.19–7.35 (m, 10H), 7.40–7.47 (m, 5H); ^13^C-NMR (100 MHz, CDCl_3_): δ = 13.9, 22.6, 27.6, 33.0, 67.7, 70.9, 86.6, 127.0 (3C), 127.8 (6C), 128.6 (6C), 143.8. 


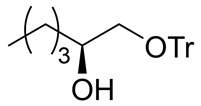


(*S*)-2-(Benzyloxy)hexan-1-ol (**8**) [15]

A mixture of benzyl bromide (4.85 mL, 41 mmol) and (*S*)-alcohol **7** (12.4 g, 34 mmol) in DMF (25 mL) were added to a stirred suspension of NaH (60%; 2.04 mg, 51 mmol) in DMF (10 mL) at 0–5 °C under an Ar atmosphere. TBAI (126 mg, 0.3 mmol) was added to the mixture and the mixture was allowed to warm up to 20–25 °C, followed by stirring for 1 h. The mixture was quenched with MeOH and K_2_CO_3_, which was extracted three times with AcOEt. The combined organic phase was washed with water, brine, dried (Na_2_SO_4_), and concentrated. The obtained crude oil (15.6 g) was used for the next step without purification.

TsOH·H_2_O (647 mg, 3.4 mmol) was added to a solution of the oil (15.6 g) in MeOH (70 mL) at 20–25 °C under an Ar atmosphere, and the mixture was stirred for 1 h at the same temperature. The mixture was quenched with sat. NaHCO_3_ aq. and concentrated, which was extracted three times with AcOEt. The combined organic phase was washed with water, brine, dried (Na_2_SO_4_), and concentrated. The obtained crude product was purified by SiO_2_–column chromatography (hexane/AcOEt = 15:1–3:1) to give **8** (6.52 g, 92% for 2 steps, >98% ee).

Yellow oil; ${\left[\alpha \right]}_{D}^{24}$ +21.4 (*c* 1.16, CHCl_3_) [lit. [15], [α${]}_{D}^{\mathrm{unknown}}$ +22.3 (*c* 1.13, CHCl_3_)]; ^1^H-NMR (400 MHz, CDCl_3_): δ = 0.90 (t, *J* = 6.9 Hz, 3H), 1.23–1.40 (m, 4H), 1.44–1.71 (m, 2H), 1.93 (brs, 1H), 3.47–3.58 (m, 2H), 3.65–3.75 (m, 1H), 4.54 (d, *J* = 11.5 Hz, 1H), 4.63 (d, *J* = 11.5 Hz, 1H), 7.27–7.39 (m, 5H); ^13^C NMR (500 MHz, CDCl_3_): δ = 13.8, 22.6, 27.3, 30.3, 63.9, 71.3, 79.7, 127.4, 127.6 (2C), 128.2 (2C), 138.3. HPLC analysis (AD-H, flow rate 1.00 mL/min, solvent: hexane/2-propanol = 30/1) t_R_(racemic) = 9.33 min and 10.27 min. t_R_[(*S*)-form] = 8.95 min.


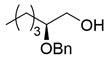


(*S*)-2-(Benzyloxy)hexanal (**1**) [15]

TEMPO (106 mg, 0.68 mmol) and KBr (407 mg, 3.4 mmol) was added to a stirred solution of alcohol **8** (7.08 g, 34 mmol) in CH_2_Cl_2_ (34 mL) at 0–5 °C under an Ar atmosphere. A mixture of NaOCl aq. (1.5 M, 34 mL, 51 mmol), NaHCO_3_ (6.7 g, 80 mmol), and Na_2_CO_3_ (318 mg, 3 mmol) in water (220 mL), was added to the solution at same temperature. The mixture was allowed to warm to 20–25 °C, followed by stirring at the same temperature for 1 h. The mixture was quenched with water, which was extracted twice with CH_2_Cl_2_. The combined organic phase was washed with water, brine, dried (Na_2_SO_4_), and concentrated. The obtained crude oil was purified by Florisil^®^ column chromatography (hexane/AcOEt = 5:1) to give the desired product **1** (6.04 g, 86%).

Yellow oil; [α${]}_{D}^{24}$ −81.2 (*c* 1.08, CHCl_3_) [lit. [15], [α${]}_{D}^{\mathrm{unknown}}$ −86.1 (*c* 0.98, CHCl_3_)]; ^1^H-NMR (500 MHz, CDCl_3_): δ = 0.90 (t, *J* = 7.5 Hz, 3H), 1.24–1.49 (m, 4H), 1.69 (q, *J* = 6.9 Hz, 13.8 Hz, 2H), 3.76 (t, *J* = 6.3 Hz, 1H), 4.54 (d, *J* = 11.5 Hz, 1H), 4.68 (d, *J* = 11.5 Hz, 1H), 7,27–7.41 (m, 5H), 9.66 (s, 1H); ^13^C NMR (125 MHz, CDCl_3_): δ = 13.7, 22.3, 26.7, 29.6, 72.3, 83.3, 127.8, 127.9, 128.4, 137.3, 203.6.

An alternative method is following:

DMSO (4.26 mL, 60 mmol) in CH_2_Cl_2_ (20 mL) was added slowly to a stirred solution of oxalyl dichloride (3.43 mL, 40 mmol) in CH_2_Cl_2_ (60 mL) at −78 °C under an Ar atmosphere. After the mixture was stirred for 5 min, **8** (4.22 g, 20 mmol) in CH_2_Cl_2_ (20 mL) was added and the mixture was stirred for 0.5 h at the same temperature. Et_3_N (16.6 mL, 120 mmol) was added to the mixture and the mixture was allowed to warm up to 0–5 °C over a period of 1 h, followed by stirring for 1 h at 0–5 °C. The mixture was quenched with water, which was extracted three times with Et_2_O. The combined organic phase was washed with a large amounts of water, brine, dried (Na_2_SO_4_), and concentrated. The obtained crude product was purified by SiO_2_–column chromatography (hexane/AcOEt = 25/1) to give the desired product **1** (3.99 g, 97%).


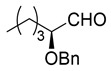


Methyl (5*S*,6*S*)-6-(benzyloxy)-5-hydroxy-3-oxodecanoate (*syn*-**3**) [15]

Preparation for Ti-BINOL solution: A suspension of Ti(O*i*Pr)_4_ (2.9 mg, 10 μmol), and (*S*)-BINOL (2.8 mg, 10 μmol) in THF (0.4 mL) was stirred stirred at 20–25 °C under an Ar atmosphere for 1 h.

Asymmetric Mukaiyama aldol reaction: Ti-BINOL solution was added to a stirred suspension of aldehyde **1** (103 mg, 0.50 mmol) and LiCl (0.85 mg, 20 μmol) in THF (0.5 mL) at 20–25 °C under an Ar atmosphere, followed by stirring at the same temperature for 0.5 h. Chan’s diene **2** (260 mg, 1.0 mmol) in THF (0.3 mL) was added slowly to the mixture, which was stirred for 14 h. PPTS (25 mg, 0.10 mmol) in MeOH (1.0 mL) was added to the mixture, followed by stirring at the same temperature for 2 h. The resulting mixture was quenched with sat. NaHCO_3_ aq., which was extracted three times with Et_2_O. The combined organic phase was washed with water, brine, dried (Na_2_SO_4_), and concentrated. The obtained crude oil was purified by SiO_2_–column chromatography (hexane/AcOEt = 8/1) to give the desired product *syn*-**3** (85% ee, dr 93:7, 51 mg, 31%).

Pale yellow oil; [α]${]}_{D}^{25}$ +1.0 (*c* 1.0, CHCl_3_) [lit. [15], [α${]}_{D}^{\mathrm{unknown}}$ +1.2 (*c* 1.00, CHCl_3_)]; 85% ee; HPLC analysis (AD-3, flow rate 1.00 mL/min, solvent: hexane/2-propanol = 30:1) t_R_(racemic) = 13.51 min, 14.13 min, 18.89 min and 19.82 min. t_R_[(5*S*,6*S*)-form] = 18.69 min. ^1^H-NMR (500 MHz, CDCl_3_): δ = 0.91 (t, *J* = 6.9 Hz, 3H), 1.24–1.70 (m, 6H), 2.62–2.64 (m, 1H), 2.71-2.74 (m, 1H), 3.34–3.37 (m, 1H), 3.477 (s, 1H), 3.480 (s, 1H), 3.73 (s, 3H), 4.13–4.18 (m, 1H), 4.49 (d, *J* = 11.5 Hz, 1H), 4.63 (d, *J* = 11.5 Hz, 1H), 7.28–7.37 (m, 5H); ^13^C-NMR (125 MHz, CDCl_3_): δ = 14.0, 22.8, 27.6, 29.3, 46.0, 49.6, 52.3, 68.3, 72.2, 80.8, 127.8, 127.9, 128.4, 138.2, 167.4, 202.7.


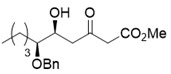


(*E*)-((1,3-dimethoxybuta-1,3-dien-1-yl)oxy)trimethylsilane (**4**) (Brassard’s diene)

Concentrated H_2_SO_4_ (0.27 mL, 5.0 mmol) was added to a stirred mixture of methyl acetoacetate (11.6g, 100 mmol) and trimethyl orthoformate (26.5 g, 250 mmol) at 0–5 °C under an Ar atmosphere, followed by stirring at 20–25 °C for 24 h. K_2_CO_3_ (5.0 g) was added to the mixture, which was filtered through a glass filter. The filtrate was concentrated under reduced pressure. The obtained crude oil was purified by distillation (bp 72–75 °C/3.2 kPa) to give the desired (*E*)-methyl-3-methoxybut-2-enoate (9.08 g, 70%).

*n*BuLi (1.63 M in hexane, 13.6 mL, 22 mmol) was added to stirred solution of *i*Pr_2_NH (3.11 mL, 22 mmol) in THF (10 mL) at 0–5 °C under an Ar atmosphere, followed by stirring for 10 min. The mixture was cooled down to −78 °C and (*E*)-methyl-3-methoxybut-2-enoate (2.22 g, 17 mmol) in THF (4.0 mL) was added to the mixture, followed by stirring at the same temperature for 0.5 h. TMSCl (2.58 mL, 20 mmol) in THF (3.0 mL) was added to the mixture at the same temperature and the mixture was allowed to warm up to 0–5 °C over a period of 1 h. The mixture was concentrated and filtered through Celite^®^ (No.503) using a glass filter, and washing with hexane (10 mL × 3). The filtrate was concentrated under reduced pressure and the obtained crude oil was purified by distillation to give the desired product **4** (2.62 g, 76%).

Colorless oil; bp 40–43 °C/50 Pa; ^1^H-NMR (400 MHz, CDCl_3_): δ = 0.26 (s, 9H), 3.56 (s, 3H), 3.57 (s, 3H), 3.99 (t, *J* = 1.4 Hz, 1H), 4.03 (d, *J* = 1.4 Hz, 1H), 4.34 (d, *J* = 1.8, 1H); ^13^C NMR (100 MHz, CDCl_3_): δ = 0.3, 54.0, 55.0, 75.5, 78.6, 158.7


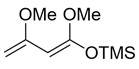


(*S*)-6-[(*S*)-1-(Benzyloxy)pentyl]-4-methoxy-5,6-dihydro-2*H*-pyran-2-one [(1′*S*,6*S*)-**5**] [15]

(1) 1M-KOH aq. (0.37 mL) was added to a stirred solution of (5*S*,6*S*)-aldol adduct *syn*-**3** (108 mg, 0.33 mmol) in MeOH (0.37 mL) at 20–25 °C under an Ar atmosphere, followed by stirring at the same temperature for 3 h. The mixture was quenched with 1M-HCl aq., which was extracted twice with AcOEt. The combined organic phase was washed with brine, dried (Na_2_SO_4_), and concentrated. The obtained crude oil was purified by SiO_2_–gel column chromatography (hexane/AcOEt = 5/1–2/1) to give the desired 4-hydroxy-5,6-dihydro-2*H*-pyran-2-one precursor (dr 93:7, 94 mg, 98%).

^1^H-NMR (500 MHz, CDCl_3_): δ = 0.92 (t, *J* = 6.9 Hz, 3H), 1.28–1.40 (m, 4H), 1.71–1.77 (m, 2H), 2.58 (dd, *J* = 5.2 Hz, 17.2 Hz, 1H), 2.76 (dd, *J* = 5.2 Hz, 17.2 Hz, 1H), 3.30 (d, *J* = 20.1 Hz, 1H), 3.41 (d, *J* = 20.1 Hz, 1H), 3.42–3.45 (m, 1H), 4.44 (d, *J* = 10.9 Hz, 1H), 4.59 (d, *J* = 10.9 Hz, 1H), 4.71–4.74 (m, 1H), 7.26–7.37 (m, 1H); ^13^C-NMR (125 MHz, CDCl_3_): δ = 13.9, 22.7, 27.5, 29.3, 40.6, 46.2, 72.3, 75.9, 80.0, 128.1, 128.2, 128.5, 136.9, 167.7, 199.4

K_2_CO_3_ (80 mg, 0.58 mmol) was added to a stirred suspension of the precursor (85 mg, 029 mmol) and Me_2_SO_4_ (55 mg, 0.44 mmol) in acetone (1.5 mL) at 20–25 °C under an Ar atmosphere, followed by stirring at the same temperature for 14 h. The mixture was quenched with water, which was extracted three times with Et_2_O. The combined organic phase was washed with brine, dried (Na_2_SO_4_), and concentrated. The obtained crude oil was purified by SiO_2_–gel column chromatography (hexane/AcOEt = 6/1–4/1) to give the desired product (1′*S*,6*S*)-**5** (dr 91:9, 79 mg, 89%).

Pale yellow oil; [α${]}_{D}^{25}$ −93.7 (*c* 0.72, CHCl_3_)]. [lit. [15], [α${]}_{D}^{\mathrm{unknown}}$ −99.1 (*c* 0.93, CHCl_3_)].

(2) TiCl_4_ (0.02 mL, 0.2 mmol) was added to a solution of aldehyde **1** (206 mg, 1.0 m mol) in CH_2_Cl_2_ (3.0 mL) at 0–5 °C under an Ar atmosphere, followed by stirring at the same temperature for 10 min. Chan’s diene (61 % purity, 520 mg, 1.2 mmol) was added to the mixture, which was stirred at 0–5 °C for 5 min and at 20–25 °C for 1 h. MeOH (2 mL) and PPTS (125 mg, 0.5 mmol) was successively added to the mixture, followed by stirring at the same temperature for 2 h. The mixture was quenched with sat. NaHCO_3_ aq., which was filtered through Celite^®^. The filtrate was extracted twice with AcOEt, and the combined organic phase was washed with water, brine dried (Na_2_SO_4_), and concentrated. The obtained crude oil was purified by SiO_2_-column chromatography (hexane/AcOEt = 4:1) to give the desired product (1’*S*,6*S*)-**5** [165 mg, 49%, 91% ee, dr = 87:13].

(3) Aldehyde **1** (413 mg, 2.0 mmol) in CH_2_Cl_2_ (1.0 mL) was added to a stirred suspension of Eu(fod)_3_ (104 mg, 0.1 mmol) in CH_2_Cl_2_ (1.0 mL) at 0–5 °C under an Ar atmosphere, followed by stirring at the same temperature for 5 min. Brassard’s diene **4** (607 mg, 3.0 mmol) in CH_2_Cl_2_ (2.0 mL) was added to the mixture at the same temperature, followed by stirring for 2 h. The mixture was quenched with water, which was extracted three times with AcOEt. The combined organic phase was washed with water, brine, dried (Na_2_SO_4_), and concentrated. The obtained crude product was purified by SiO_2_–column chromatography (hexane/AcOEt = 3/1) to give the desired product [(1’*S*,6*S*)-**5**] (370 mg, 67%, >98% ee, dr = 98:2). HPLC analysis (AD-3, flow rate 1.00 mL/min, solvent: hexane/2-propanol = 30:1) t_R_(racemic) = 23.25 min and 24.77 min. t_R_[(1*S*,6*S*)-form] = 25.53 min.; ^1^H-NMR (500 MHz, CDCl_3_): δ = 0.89 (t, *J* = 6.9 Hz, 3H), 1.25–1.72 (m, 6H), 2.26 (dd, *J* = 4.0 Hz, 17.2 Hz, 1H), 2.70 (ddd, *J* = 1.7 Hz, 13.2 Hz, 17.2 Hz, 1H), 3.58–3.61 (m, 1H), 3.74 (s, 3H), 4.52 (dt, *J* = 4.0 Hz, 13.2 Hz, 1H), 4.62 (d, *J* = 11.5 Hz, 1H), 4.66 (d, *J* = 11.5 Hz, 1H), 5.13 (d, *J* = 1.7 Hz, 1H), 7.27–7.36 (m, 5H); ^13^C NMR (125 MHz, CDCl_3_): δ = 14.0, 22.7, 27.9, 28.4, 29.3, 56.1, 72.9, 76.3, 79.0, 90.2, 127.8, 127.9, 128.4, 138.1, 167.0, 173.3. 


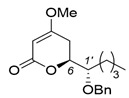


(−)-Pestalotin; (*S*)-6-[(*S*)-1-Hydroxypentyl]-4-methoxy-5,6-dihydro-2*H*-pyran-2-one [9]

A suspension of benzyl ether [(1*S*,6*S*)-**5**] (448 mg, 1.5 mmol) and 20% Pd(OH)_2_/C (53 mg, 0.08 mmol) in AcOEt (15 mL), equipped with a H_2_ balloon, was stirred at 20–25 °C for 1 h. The mixture was filtered through Celite^®^ (No.503) using glass filter and the filtrate was concentrated under reduced pressure. The obtained crude solid (384 mg) was purified by SiO_2_–column chromatography (hexane/AcOEt = 3:2) to give the desired (−)-pestalotin (283 mg, 88%, >98% ee, dr = >98:2).

Colorless crystals; mp 84–86 °C (lit. [9], 85.8–86.0 °C); [α${]}_{D}^{25}$ −91.9 (*c* 0.44, MeOH) [lit. [9], [α${]}_{D}^{21}$ −90.2 (*c* 1.17, MeOH)]; HPLC analysis (AD-3, flow rate 1.00 mL/min, solvent: hexane/2-propanol = 30:1) t_R_(racemic) = 45.23 min and 48.13 min. t_R_[(1*S*,6*S*)-form] = 45.85 min.; ^1^H-NMR (500 MHz, CDCl_3_): δ = 0.92 (t, *J* = 6.9 Hz, 3H), 1.30–1.67 (m, 6H), 2.07 (brs, 1H), 2.25 (dd, *J* = 4.0 Hz, 17.2 Hz, 1H), 2.80 (ddd, *J* = 1.7 Hz, 12.6 Hz, 17.2 Hz, 1H), 3.61–3.64 (m, 1H), 3.76 (s, 3H), 4.30 (dt, *J* = 4.0 Hz, 12.6 Hz, 1H), 5.15 (d, *J* = 1.7 Hz, 1H); ^13^C-NMR (125 MHz, CDCl_3_): δ = 13.9, 22.6, 27.6, 29.6, 32.4, 56.1, 72.4, 78.4, 90.0, 166.7, 173.1.


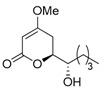


(*R*)-6-[(*S*)-1-(Benzyloxy)pentyl]-4-methoxy-5,6-dihydro-2*H*-pyran-2-one [(1’*S*,6*R*)-5]

(1) Aldehyde **1** (206 mg, 1.0 mmol) was added to a stirred suspension of ZrCl_4_ (47 mg, 0.2 mmol) in CH_2_Cl_2_ (0.9 mL) at 0–5 °C under an Ar atmosphere. After 10 min, Chan’s diene (ca. 60% purity; 520 mg, 1.2 mmol) was added to the mixture, which was allowed to warm up to 20–25 °C, followed by stirring for 1 h. MeOH (2.0 mL) and PPTS (125 mg 0.5 mmol) was successively added to the solution, followed by stirring at 40–45 °C for 14 h. Sat. NaHCO_3_ aq. solution was added to the mixture, which was filtered through Cerite^®^. The filtrate was extracted twice with AcOEt, and the combined organic phase was washed with water, brine dried (Na_2_SO_4_), and concentrated. The obtained crude oil was purified by SiO_2_-column chromatography (hexane/AcOEt = 4:1) to give the desired product (1′*S*,6*R*)-**5** (126 mg, 41%, >98% ee, dr = 35:65).

Colorless oil. HPLC analysis (AD-3, flow rate 1.00 mL/min, solvent: hexane/2-propanol = 30:1) t_R_(racemic) = 17.72 min and 19.60 min. t_R_[(1*S*,6*R*)-form] = 18.10 min.; ^1^H-NMR (500 MHz, CDCl_3_): δ = 0.89 (t, *J* = 6.9 Hz, 3H), 1.29-1.64 (m, 6H), 2.35 (dd, *J* = 4.0 Hz, 17.2 Hz, 1H), 2.81 (ddd, *J* = 1.7 Hz, 12.6 Hz, 17.2 Hz, 1H), 3.74 (s, 3H), 3.73-3.78 (m, 1H), 4.39 (dt, *J* = 4.0, 12.6, 1H), 4.63 (d, *J* = 11.5, 1H), 4.74 (d, *J* = 11.5, 1H), 5.14 (d, *J* = 1.7, 1H), 7.28-7.35 (m, 5H ); ^13^C-NMR (125 MHz, CDCl_3_): δ = 13.9, 22.6, 27.4, 27.9, 30.7, 56.0, 73.3, 78.4, 79.1, 90.0, 127.6, 127.8, 128.3, 138.3, 167.0, 173.4.

(2) Et_2_AlCl (1.0 M, 0.6 mL, 0.6 mmol) was added to a stirred solution of aldehyde **1** (103 mg, 0.5 mmol) in CH_2_Cl_2_ (0.5 mL) at −78 °C under an Ar atmosphere. After 5 min, diene **4** (202 mg, 0.6 mmol) in CH_2_Cl_2_ (0.5 mL) was added to the mixture, which was stirred for 14 h at the same temperature. The mixture was allowed to warm up to −30 °C, followed by stirring for 14 h. The mixture was quenched by MeOH, which was extracted three times with AcOEt. The combined organic phase was washed with water, brine, dried (Na_2_SO_4_), and concentrated. The obtained crude product was purified by SiO_2_-column chromatography (hexane/AcOEt = 5/1) to give a mixture of aldol adduct **9** and (1′*S*,6*R*)-**5** (45:55, 48 mg, 30%). The mixture (48 mg) and PPTS (2 mg, 0.007 mmol) in toluene (1.4 mL), was added at 80–85 °C for 1 h under an Ar atmosphere. After cooling to room temperature, water was added to the mixture, which was extracted twice with AcOEt. The combined organic phase was washed with water and brine, dried (Na_2_SO_4_), and concentrated. The obtained crude solid purified by SiO_2_-column chromatography (hexane/AcOEt = 5/1) to give the desired product (1’*S*,6*R*)-**5** (23 mg, 2 steps 15%, ca. 30% of (1’*S*,6*S*)-**5** was contained).


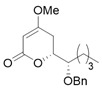


(+)-Epipestalotin; (*R*)-6-[(*S*)-1-Hydroxypentyl]-4-methoxy-5,6-dihydro-2*H*-pyran-2-one [9]

A suspension of benzyl ether [(1*S*,6*R*)-**5**] (365 mg, 1.2 mmol) and 20% Pd(OH)_2_/C (42 mg, 0.06 mmol) in AcOEt (12 mL), equipped with a H_2_ balloon, was stirred stirred at 20–25 °C for 1 h. The mixture was filtered through Celite^®^ (No.503) using glass filter and the filtrate was concentrated under reduced pressure. The obtained crude solid was purified SiO_2_–column chromatography (hexane/AcOEt = 3/2) to give the desired (+)-epipestalotin (187 mg, 71%, >98% ee, dr = 98:2). 

Colorless crystals; mp 92–94 °C (lit. [9], 93.0–94.0 °C); [α${]}_{D}^{20}$ + 75.3 (*c* 0.39, MeOH) [lit. [9], [α${]}_{D}^{17}$ + 75.9 (*c* 0.39, MeOH)]; >99% ee; HPLC analysis (AD-3, flow rate 1.00 mL/min, solvent: hexane/2-propanol = 25:1) t_R_(racemic) = 33.00 min and 35.46 min. t_R_[(1*S*,6*R*)-form] = 34.81 min.; ^1^H NMR (500 MHz, CDCl_3_): δ = 0.92 (t, *J* = 6.9 Hz, 3H), 1.30–1.56 (m, 6H), 2.04 (brs, 1H), 2.24 (dd, *J* = 4.0 Hz, 17.2 Hz, 1H), 2.84 (ddd, *J* = 1.7 Hz, 12.6 Hz, 17.2 Hz, 1H), 3.76 (s, 3H), 3.94–3.97 (m, 1H), 4.34 (dt, *J* = 3.4 Hz, 12.6 Hz, 1H), 5.14 (d, 1.7 Hz, 1H); ^13^C NMR (125 MHz, CDCl_3_): δ = 13.8, 22.4, 26.8, 27.7, 31.4, 56.0, 71.3, 78.7, 89.7, 167.1, 173.5. 


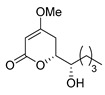


(−)-Epipestalotin; (*S*)-6-[(*R*)-1-Hydroxypentyl]-4-methoxy-5,6-dihydro-2*H*-pyran-2-one [9]

DEAD (40% in toluene, 0.91 mL, 2.0 mmol) was added slowly to a stirred mixture of (−)-pestalotin (214 mg, 1.0 mmol) and 4-nitrobenzoic acid (334 mg, 2.0 mmol) and PPh_3_ (525 mg, 2.0 mmol) in toluene (10 mL) at 0–5 °C under an Ar atmosphere, followed by stirring at 20–25 °C for 6 h. The mixture was quenched with water, which was extracted three times with AcOEt. The combined organic phase was washed with sat. NaHCO_3_ aq., brine, dried (Na_2_SO_4_), and concentrated. The obtained crude product was purified by SiO_2_–column chromatography (hexane/AcOEt = 3:1) to give a mixture of the desired (−)-Epipestalotin and diethyl hydrazodicarboxylate, which was used in the next step without further purification. 

A suspension of the mixture and K_2_CO_3_ (138 mg, 1.0 mmol) in MeOH (10 mL) was stirred at 20–25 °C under an Ar atmosphere for 10 min. The mixture was filtered through Celite^®^ (No.503) using a glass filter washing with AcOEt (5 mL × 3). The filtrate was concentrated under reduced pressure and the obtained crude oil, which was purified by SiO_2_–column chromatography (hexane/AcOEt = 2:1) to give the desired (−)-epipestalotin (133 mg, 62% for 2 steps, >98% ee, dr = >98:2). 

Colorless crystals; mp 89–91 °C (lit. [9], 90.7–91.2 °C); [α${]}_{D}^{20}$ −75.8 (*c* 0.58, MeOH) [lit. [9], [α${]}_{D}^{17}$ −75.6 (*c* 0.58, MeOH)]; HPLC analysis (AD-3, flow rate 1.00 mL/min, solvent: hexane/2-propanol = 25:1) t_R_(racemic) = 33.00 min and 35.46 min. t_R_[(1*R*,6*S*)-form] = 32.31 min.; ^1^H-NMR (500 MHz, CDCl_3_): δ = 0.92 (t, *J* = 6.9 Hz, 3H), 1.30–1.55 (m, 6H), 2.04 (brs, 1H), 2.24 (dd, *J* = 4.0 Hz, 17.2 Hz, 1H), 2.84 (ddd, *J* = 1.7 Hz, 12.6 Hz, 17.2 Hz, 1H), 3.76 (s, 3H), 3.94–3.97 (m, 1H), 4.34 (dt, *J* = 3.4 Hz, 12.6 Hz, 1H), 5.14 (d, *J* = 1.7 Hz, 1H); ^13^C-NMR (125 MHz, CDCl_3_): δ = 13.8, 22.4, 26.8, 27.7, 31.4, 56.0, 71.3, 78.7, 89.7, 167.1, 173.5.


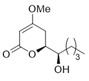


(+)-Pestalotin; (*R*)-6-[(*R*)-1-Hydroxypentyl]-4-methoxy-5,6-dihydro-2*H*-pyran-2-one [9]

Following the procedure for the preparation of (−)-epipestalotin, the reaction of (+)-epipestalotin (107 mg, 0.5 mmol) using DEAD (40% in toluene, 0.45 mL, 1.0 mmol), 4-nitrobenzoic acid (167 mg, 1.0 mmol), PPh_3_ (262 mg, 1.0 mmol), and K_2_CO_3_ (69 mg, 0.5 mmol) give the desired (+)-pestalotin (72 mg, 67% for 2 steps, >98% ee, dr > 98:2,).

Colorless crystals; mp 82–84 °C (lit. [9], 83.0–84.5 °C); [α${]}_{D}^{20}$ +97.5 (*c* 0.65, MeOH) [lit. [9], [α${]}_{D}^{17}$ +88.7 (*c* 0.65, MeOH)]; HPLC analysis (AD-3, flow rate 1.00 mL/min, solvent: hexane/2-propanol = 30:1) t_R_(racemic) = 45.23 min and 48.13 min. t_R_[(1*R*,6*R*)-form] = 49.57 min; ^1^H-NMR (500 MHz, CDCl_3_): δ = 0.92 (t, *J* = 6.9 Hz, 3H), 1.30–1.67 (m, 6H), 2.07 (brs, 1H), 2.25 (dd, *J* = 4.0 Hz, 17.2 Hz, 1H), 2.80 (ddd, *J* = 1.7 Hz, 12.6 Hz, 17.2 Hz, 1H), 3.61–3.64 (m, 1H), 3.76 (s, 3H), 4.30 (dt, *J* = 4.0 Hz, 12.6 Hz, 1H), 5.15 (d, *J* = 1.7 Hz, 1H); ^13^C-NMR (125 MHz, CDCl_3_): δ = 13.9, 22.6, 27.6, 29.6, 32.4, 56.1, 72.4, 78.4, 90.0, 166.7, 173.1. 


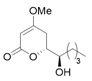


## 4. Conclusions 

We achieved an asymmetric total synthesis of all four chiral pestalotin diastereomers using common and commercially-available (*R*)-glycidol as the starting compound. The present synthesis involves a couple of divergent strategies, including syn- and anti-selective Mukaiyama aldol additions and hetero-Diels-Alder reactions. 

Catalytic asymmetric Mukaiyama aldol reactions of readily-available *bis*(TMSO)diene (Chan’s diene) with (*S*)-2-benzyloxyhexanal derived from (*R*)-glycidol afforded a *syn*-aldol adduct with high diastereoselectivity and enantioselectivity. Diastereoselective Mukaiyama aldol reactions mediated by catalytic achiral Lewis acids directly produced not only a (1′*S*,6*S*)-pyrone precursor via the *syn*-aldol adduct using TiCl_4_, but also (1′*S*,6*R*)-pyrone precursor derived from an antialdol adduct using ZrCl_4_ in a stereocomplementary manner. 

A hetero-Diels-Alder reaction of similarly available mono(TMSO)diene (Brassard’s diene) with (*S*)-2-benzyloxyhexanal produced the (1’*S*,6*S*)-pyrone precursor promoted by Eu(fod)_3_ and the (1′*S*,6*R*)-pyrone precursor EtAlCl_2_. 

Debenzylation of (1′*S*,6*S*)-and (1′*S*,6*R*)-precursors furnished natural (−)(−)-pestalotin and unnatural (+)-epipestalotin, respectively. The unnatural (+)-pestalotin and (−)-epipestalotin were successfully synthesized by Mitsunobu inversion of (−)-pestalotin and (+)-epipestalotin, respectively, in a divergent manner. All four chiral pestalotin diastereomers obtained possessed high chemical and optical purities (optical rotations, ^1^H-NMR, ^13^C-NMR, and HPLC measurements). 

The present divergent method affords concise access to asymmetric syntheses directed for these types of compounds with consecutive chiral dihydroxy groups, and is useful for accessible asymmetric Mukaiyama aldol reactions and relevant hetero-Diels-Alder reactions.

Copies of the ^1^H, ^13^C-NMR spectra for compounds *syn*-**3**, (1′*S*,6*S*)-**5**, (−)-pestalotin, (1′*S*,6*R*)-**5**, (+)-epipestalotin are available in the Appendix A. Copies of the HPLC chromatogram of (±)-**8**, (*S*)-**8**, (±)-**3**, syn-**3**, (±)-**5**, (1′S,6S)-**5**, (1’S,6R)-**5**, (±)-pestalotin, (+)-pestalotin, (−)-pestalotin, (±)-epipestalotin, (+)-epipestalotin, and (−)-epipestalotin are available in the Appendix A.

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
