# Peer review of "Divergent Asymmetric Total Synthesis of All Four Pestalotin Diastereomers from (R)-Glycidol"

_molecules, 2020, doi:10.3390/molecules25020394_

Round 1
Reviewer 1 Report
This manuscript by Tanabe describes a divergent asymmetric approach towards Pestalotin Families from common precursor (R)-Glycidol. The key reaction ire represented by the Mukaiyama aldol reaction as well as the analogous formal hetero-Diels-Alder reaction that delivers the key intermediate in a diastereoselective manner. This kind of divergent approach from a single precursor is unprecedented. Also, various conditions were examined for the key transformation. Based upon this merit, publication of this manuscript in Molecules is recommended, after the authors address the following issues in the revised manuscript.
In the title: The word “Asymmetric” and “Chiral” looks redundant.
In line 38: “Asymmetric total synthesis” seems more appropriate.
In line 103: “TMPO” should be changed to “TEMPO”
In line 105: Does this 4-step yield include TEMPO oxidation or Moffatt-Swern oxidation?
In Scheme 2: The result of Moffatt-Swern oxidation should be described in the scheme.
p.s: I’m not a big fan of step-counting for the total synthesis. But, I suggest the authors re-count the number of steps. (For example, the author claimed five steps for 1. The key reaction should cost one more step. The debenzylation completes the total synthesis. Thus, it appears the synthesis of Pedestalotin requires at least 7 (longest linear) steps. In addition, the Mitsunobu inversion adds “two” more steps because the intermediate is obtained as isolable entity).
Author Response
Comments and Suggestions for Authors
This manuscript by Tanabe describes a divergent asymmetric approach towards Pestalotin Families from common precursor (R)-Glycidol. The key reaction ire represented by the Mukaiyama aldol reaction as well as the analogous formal hetero-Diels-Alder reaction that delivers the key intermediate in a diastereoselective manner. This kind of divergent approach from a single precursor is unprecedented. Also, various conditions were examined for the key transformation. Based upon this merit, publication of this manuscript in Molecules is recommended, after the authors address the following issues in the revised manuscript.
>In the title: The word “Asymmetric” and “Chiral” looks redundant.
Yes, “Chiral” was deleted.
>In line 38: “Asymmetric total synthesis” seems more appropriate.
Yes, I altered it.
>In line 103: “TMPO” should be changed to “TEMPO”
Yes, I altered it.
>In line 105: Does this 4-step yield include TEMPO oxidation or Moffatt-Swern oxidation?
This is TEMPO oxidation, which is more accessible procedure. The data using Moffatt-Swern oxidation was added in the parenthesis.
>In Scheme 2: The result of Moffatt-Swern oxidation should be described in the scheme.
Yes, I added it.
>p.s: I’m not a big fan of step-counting for the total synthesis. But, I suggest the authors re-count the number of steps. (For example, the author claimed five steps for 1. The key reaction should cost one more step. The debenzylation completes the total synthesis. Thus, it appears the synthesis of Pedestalotin requires at least 7 (longest linear) steps. In addition, the Mitsunobu inversion adds “two” more steps because the intermediate is obtained as isolable entity).
Thank you for the reviewer’s reasonable suggestion. I carefully re-count the number of steps for 1 and this is really 5 steps. On the other hand, for the synthesis of (-)-pestalotin and (+)-epipestalotin I altered it from 6 steps to 7 steps, and also for (-)-epipestalotin and (+)-pestalotin, from 7 steps to 9 steps.
Reviewer 2 Report
The manuscript describes a stereodivergent synthesis of four diastereomers of pestalotin from one common precursor from a chiral pool, namely (R)-glycidol. The presented protocols are based on previously used methods (asymmetric Mukaiyama aldol reaction or hetero-Diels-Alder reaction), but the numer of steps and total yield are improved. The manuscrpt is written in good scientific language, the presentation is logical and conclusions justified.
My only concerns are as follows:
1. The word "famielies" used in the title and throughout the text suggested to me that not only isomers of pestalotin, but also their derivatves would be prepared. I suggest changing to "diastereomers".
2. In the experimental part, the Authors describe their instrumentation used for measurement of IR and mass spectra. However, I could not find any data of this kind. In particular, I would expect the results of high resolution mass spectrometry at least for new compounds to be given.
Author Response
Comments and Suggestions for Authors
The manuscript describes a stereodivergent synthesis of four diastereomers of pestalotin from one common precursor from a chiral pool, namely (R)-glycidol. The presented protocols are based on previously used methods (asymmetric Mukaiyama aldol reaction or hetero-Diels-Alder reaction), but the numer of steps and total yield are improved. The manuscrpt is written in good scientific language, the presentation is logical and conclusions justified.
My only concerns are as follows:
>1. The word "famielies" used in the title and throughout the text suggested to me that not only isomers of pestalotin, but also their derivatves would be prepared. I suggest changing to "diastereomers".
Yes, I changed "families" to “diastereomers” in all sentences.
>2. In the experimental part, the Authors describe their instrumentation used for measurement of IR and mass spectra. However, I could not find any data of this kind. In particular, I would expect the results of high resolution mass spectrometry at least for new compounds to be given.
Thank you for your comment. I deleted the IR apparatus.